# Exploratory study of serum protein biomarkers for sudden cardiac arrest using protein extension assay: A case-control study

Ha Yeon Shin[1], Jeong Ho Park[2,3,4*], Kyoung-Chul Cha[5,6*], Hyun Je Kim[1,7,8], Woo Jin Jung[5,6], Seulki Choi[2,3,4], Ji Hwan Moon[9], Young Il Roh[5,6], Young Sun Ro[2,3,4], Sung Oh Hwang[5,6], Sang Do Shin[2,3,4], for CAPTURES-II Investigators[¶]

1 Department of Biomedical Sciences, Seoul National University Graduate School, Seoul, Republic of Korea, 2 Department of Emergency Medicine, Seoul National University College of Medicine and Hospital, Seoul, Republic of Korea, 3 Laboratory of Emergency Medical Services, Seoul National University Hospital Biomedical Research Institute, Seoul, Republic of Korea, 4 Disaster Medicine Research Center, Seoul National University Medical Research Center, Seoul, Republic of Korea, 5 Department of Emergency Medicine, Yonsei University Wonju College of Medicine, Wonju, Republic of Korea, 6 Research Institute of Resuscitation Science, Yonsei University Wonju College of Medicine, Wonju, Republic of Korea, 7 Department of Microbiology and Immunology, Seoul National University College of Medicine, Republic of Korea, 8 Genomic Medicine Institute, Seoul National University College of Medicine, Republic of Korea, 9 Samsung Genome Institute, Samsung Medical Center, Seoul, Republic of Korea

¶ Membership of The CAPTURES-II Investigators Team is provided in the Acknowledgments.
* chaemp@yonsei.ac.kr; timthe@gmail.com

## Abstract

### Background

Biomarkers associated with the occurrence of sudden cardiac arrest (SCA) are not currently utilized in clinical practice. We aimed to identify novel protein biomarkers associated with sudden cardiac arrest (SCA) using proteomic profiling and evaluate their predictive power alongside traditional cardiovascular risk factors.

### Methods

A total of 42 SCA patients with medical causes, aged ≤65 years and whose initial rhythm was shockable, and 42 age- and sex-matched controls were analyzed. The initial serum samples obtained after emergency department visits were used for SCA cases. Using a protein extension assay, we identified significant biomarkers through correlation analysis with SCA and extracted proteins with no or weak correlation with the initial lactate level and arrest-to-sampling time to account for post-cardiac arrest changes. The area under the receiver operating characteristic curve (AUROC) was calculated to assess the predictive performance of the extracted proteins.

### Results

Among the 246 distinct proteins that met quality criteria, 97 showed a strong correlation with SCA. Among these 97 proteins, 44 showed weak or no correlation with lactate levels, and 12 showed weak or no correlation with onset-to-sampling time. Two proteins (AXL

**Data availability statement:** The Korea Disease Control and Prevention Agency has authority for all of the CAPTURES project, and Data cannot be shared publicly because of consent of personal information. Permission was required to use the dataset; requests can be sent for access permission to CAPTURES datasets (email: jhjeong1107@korea.kr).

**Funding:** This study was supported by the Research Program funded by the Korea Disease Control and Prevention Agency (Grant No: 2017NE3300600, 2017E3300601, 2019P330800, 2021-12-202, 2022-12-204). The funders had no role in study design, data collection and analysis, decision to publish, or preparation of the manuscript.

**Competing interests:** The authors have declared that no competing interests exist.

receptor tyrosine kinase [AXL] and TIMP Metallopeptidase inhibitor 4 [TIMP-4]) met all the criteria for biomarker extraction. Both showed significant associations with SCA and enhanced predictive power when combined with traditional risk factors in multivariable analysis. The AUROC for the baseline model using traditional risk factors was 0.692 (95% confidence interval [CI] 0.578–0.806), which improved significantly with the addition of AXL and TIMP-4 (AUROC [95% CI] 0.891 [0.817–0.964] and 0.910 [0.910–0.997], respectively).

## Conclusion

AXL and TIMP-4 may be crucial role in the early detection and risk assessment of SCA. Future research to verify the utility of AXL and TIMP-4 in large cohorts is warranted.

## Introduction

Sudden cardiac arrest (SCA) is a significant health burden worldwide, and the survival rate has remained low for decades [1]. The annual incidence of SCA is estimated to be 62 per 100,000 in the USA, 50–90 per 100,000 in European countries, and 40–90 per 100,000 in Asian countries [2]. Because of the high fatality rate, identifying high-risk populations as candidates for preventive interventions is crucial for reducing its burden of disease [3]. Currently, no widely utilized methods exist in clinical practice for detecting these high-risk populations. Various biomarkers, including genomic, proteomic, clinical and, imaging biomarkers, have been investigated in previous studies [4–9]. However, these studies have various limitations, including low accuracy, limited sample size, and targeting of specific disease groups [4,8].

Compared to other biomarkers, protein biomarkers are more cost-effective, provide rapid results, and enable real-time monitoring of disease states and treatment responses [10]. Previous proteomic biomarker studies for SCA occurrence are often limited by their focus on commonly used biomarkers like C-reactive protein (CRP) and N-terminal pro-B-type natriuretic peptide (NT-proBNP), and by often relying on samples collected long before or long after the cardiac event occurred [4,5,8]. Recently, a study utilizing mass spectrometry proteomics analysis was conducted on 330 proteins using samples from SCA survivors and age- and gender-matched control groups, resulting in the proposal of 26 new protein biomarkers [8]. However, many well-known proteins related to cardiovascular diseases were missed in the exploration, and the analysis included only survivors, with samples collected a median of 11 months after SCA. To address these limitations and improve the understanding of protein biomarkers in SCA, further research is needed to assess well-known cardiovascular-related proteins in more timely collected samples, ideally from both survivors and non-survivors, to enhance the accuracy and relevance of biomarker discovery.

The aim of this study was to conduct an exploratory analysis to elucidate the association between well-recognized proteins related to cardiovascular, inflammatory and immune diseases and the occurrence of SCA using protein extension assay technique, [11] and to evaluate their predictive power alongside traditional cardiovascular risk factors.

## Methods

### Study design

This case-controlled study is a part of the Cardiac Arrest Pursuit Trial with Unique Registration and Epidemiologic Surveillance (CAPTURES) project in Korea [12]. This study aimed

to identify the risk factors of SCA and develop preventive strategies against it, and has been ongoing since September 29, 2017. In this study, data from 17 participating hospitals from September 29, 2017 to April 30, 2022 were analysed.

## Ethics statements

The study was approved by the ethics committees of all participating centers (S1 Table). All participants or their proxy provided written informed consent before taking part in the study and the study complied with the tenets of the Declaration of Helsinki. This study is registered at ClinicalTrials.gov (NCT03700203). No minors were included in the study.

## Populations

SCA patients aged 20–79 years, who experienced cardiac arrest due to medical causes and were treated by emergency medical services before arrival at the emergency department (ED), were enrolled in the CAPTURES project. Patients with terminal illnesses, pregnancies, in hospice care, living alone, homeless, without reliable information sources, or with a 'Do Not Resuscitate' card were excluded.

Among the enrolled patients, only SCA patients aged ≤ 65 years whose initial rhythm was shockable in the ED were included in this study because we wanted to focus primarily on relatively young SCA patients with a shockable rhythm and exclude patients with a long lapse of time from the cardiac arrest to ED arrival.

Community-based voluntary controls were enrolled from two centers representing metropolitan and non-metropolitan areas. All controls were recruited in collaboration with public health centers or community centers where the project was promoted. One or two controls matched for age, sex, and urbanization level of residence were recruited in each case.

## Sample collection

Structured questionnaires, physical examination, routine laboratory analysis, and blood sampling were conducted for the patients and controls Blood samples (20 mL) were drawn and split into an EDTA tube and two serum-separating tubes (SSTs). The SSTs were centrifuged within 2 hours of sampling. After refrigerated storage of blood samples at 2–8 °C, all blood samples were sent to an external laboratory (Seoul Clinical Laboratories, Seoul, Republic of Korea) for storage and future study. Blood samples were sent to the laboratory once daily on weekdays. For the patients, blood sample extraction was recommended during the initial management, but blood samples collected within 24 h of the ED visit were also included in the study. Lactate was included in the routine laboratory analysis for patients.

## Protein analysis

We used three Olink target panels: the Cardiovascular II (version 5007) 96-Plex panel, Cardiovascular III (version 6114) 96-Plex panel, and Immuno-Oncology (version 3113) panel. The Cardiovascular panels cover proteins associated with biological functions linked to cardiovascular and inflammatory diseases, while the Immuno-Oncology panel covers proteins associated with cancer, the immune system, and systemic inflammation. Serum protein levels were measured using proximity extension immunoassay (PEA) (Olink Proteomics, Uppsala, Sweden). The Olink PEA technology uses a dual-recognition DNA-coupled immunoassay that rapidly allows for protein identification with high sensitivity and specificity. Proteomic level assessments have been described in detail previously [11]. Protein levels were measured on a relative scale and presented as normalised protein expression (NPX), which is an arbitrary

unit on a log2 scale. A high NPX value corresponds to a high protein concentration. The levels of different proteins cannot be compared using NPX. However, using inter-plate controls (IPC), any systematic differences across different plates were adjusted; therefore, a consistent comparison of the same protein levels across different plates was possible. The IPC consists of a pool of 92 antibodies, each with unique DNA-tags, and is included in triplicate on each plate. The IPC serves as a synthetic sample, expected to give a high signal across all assays, and the median of the IPC triplicates is used to normalize each assay, correcting for potential variation between runs and plates. Except for the use of IPC in triplicate, the samples themselves were not measured in replicate. Each panel analyzed 92 proteins, totaling 276 proteins with 18 overlapping. NPX values from the panel with the fewest quality control flags were kept for overlaps. The limit of detection (LOD) of each protein was estimated based on the concentration in the negative controls in each sample plate. Proteins were excluded if more than 25% of the measurements were below the LOD. For the remaining proteins, the values below the LOD were replaced by the respective LOD. Since the samples were randomized before analysis, the placement of case and control samples within the plate was random.

## Statistical analysis

All analyses were performed using the R environment for statistical computing, version 4.2.1.

The association between proteins and SCA was assessed using a two-sided rank-based Spearman test. We labeled SCA as 1 and the control as 0 and calculated Spearman's correlation coefficient for each protein. Power analysis was carried out, and we have 0.95 power at 0.05 significance level to detect correlation of 0.591. Therefore, proteins with a correlation exceeding the cutoff (|Spearman's correlation coefficient|>0.591) were extracted. Among them, we further extracted proteins with low post-cardiac arrest changes, because SCA causes systemic ischemia and inflammation, which affect the levels of various proteins. The procedure was performed in two steps. First, from the 40 SCA patients with confirmed lactate levels, we extracted proteins with no or weak correlation between lactate and protein levels (|Spearman's correlation coefficient|<0.1). The lactate level is a sensitive marker of cellular hypoxia, including cardiac arrest [13,14]. Second, from the 20 SCA patients with confirmed arrest time and blood sampling performed within 60 min of SCA onset, we extracted proteins with no or weak correlation between onset-to-sampling time and protein level (|Spearman's correlation coefficient|<0.1). A full list of proteins and the results of each extraction step are available in S2 and S3 Tables.

The distribution of extracted proteins according to SCA was plotted using boxplot, and t-test was used to compare protein levels between groups. A heatmap was also used to visualize the distribution of biomarkers, and a hierarchical cluster analysis was performed. The heatmap reorders the rows and columns of the dataset to place data with similar profiles close to one another. Subsequently, ranges of similar values were assigned specific color codes, and each entry in the data matrix was displayed graphically as one specific colour according to its degree of expression. We also performed a Gene Ontology (GO) Slim summary to simplify the interpretation of the gene ontology analysis [15]. GO is a widely used bioinformatics tool that provides a standardised vocabulary for describing genes and their products [16,17]. GO Slim is a subset of the full GO dataset, which includes a number of terms selected from each of the three main GO categories (biological process, molecular function, and cellular component). For exploratory analysis, we plotted the relationship between age, BNP, and protein levels in the SCA group and control groups using a scatterplot and smooth line with a fitted linear line since age is an important demographic factor and BNP is a well-known risk factor for cardiac arrest [4]. To evaluate predictive performance of extracted proteins, we calculated area under the receiver operating characteristic curve (AUROC). Multivariable logistic regression models

were constructed using extracted proteins, with six traditional risk factors (age, sex, diabetes, hypertension, myocardial infarction, stroke) included as independent variables. Multivariable logistic regression models were also constructed based on BNP, with traditional risk factors and extracted proteins added separately. DeLong's test was utilized to assess whether there is a statistically significant difference between each ROC curve of the models [18].

## Results

### Demographic findings

During the study period, a total of 1,228 SCA patients and 2,065 controls were enrolled in CAPTURES project. Among the SCA cases included, 60 patients aged ≤65 years with a shockable initial rhythm at the ED were identified. Among the 60 SCA patients with a shockable rhythm, a random sample of 42 patients was analyzed with 42 matched controls. For the patient group, 42 cases were collected from 13 centers as follows: 5, 4, 7, 5, 2, 2, 1, 2, 5, 1, 4, 1, and 3 cases, respectively. For the control group, 26 and 10 cases were collected from two centers, respectively.

The demographic characteristics of the 42 patients and 42 controls are shown in Table 1. Each group had 35 males (83.3%) with a median (IQR) age of 56 (60–61) years. The number of comorbidities was significantly higher in the cases (Table 1).

Among 42 cases, survival to admission and survival to discharge were 36 (85.7%) and 23 (54.8%), respectively. Coronary angiography was performed in 23 (54.8%) cases, and percutaneous coronary intervention was performed in 13 (31.0%) cases. Among the 35 patients

Table 1. Demographic characteristics of the study population.

| | Case | | Control | | |
|---|---|---|---|---|---|
| | N | % | N | % | *P*-value |
| Total | 42 | | 42 | | |
| Male | 35 | 83.3 | 35 | 83.3 | 1 |
| Age group, years | | | | | 1 |
| 20–29 | 1 | 2.4 | 1 | 2.4 | |
| 30–39 | 2 | 4.8 | 2 | 4.8 | |
| 40–49 | 6 | 14.3 | 6 | 14.3 | |
| 50–59 | 18 | 42.9 | 8 | 19.0 | |
| 60–65 | 15 | 35.7 | 10 | 23.8 | |
| Median (IQR) | 56 (50–61) | | 56 (50–61) | | |
| Metropolitan | 26 | 61.9 | 26 | 61.9 | 1 |
| Comorbidities | | | | | |
| Diabetes | 11 | 26.2 | 5 | 11.9 | 0.096 |
| Hypertension | 18 | 42.9 | 9 | 21.4 | 0.036 |
| Arrhythmia | 3 | 7.1 | 3 | 7.1 | 1 |
| Stroke | 3 | 7.1 | 1 | 2.4 | 0.616 |
| Acute myocardial infarction | 3 | 7.1 | 0 | 0.0 | 0.241 |
| Heart failure | 5 | 11.9 | 0 | 0.0 | 0.055 |
| Number of comorbidities | | | | | 0.016 |
| 0 | 16 | 38.1 | 29 | 69.0 | |
| 1 | 14 | 33.3 | 8 | 19.0 | |
| 2–6 | 12 | 28.6 | 5 | 11.9 | |

IQR, interquartile range.

with confirmed arrest time and blood sampling time, the median arrest-to-sampling time was 55 minutes (IQR: 35–105). Among the 40 patients with confirmed initial lactate levels, the median lactate level was 11.3 mmol/L (IQR: 10.1–14.7).

## Biomarker extraction and exploration

Among the 258 distinct proteins, 12 proteins were excluded for analysis because more than 25% of the measurements were below the LOD. Of the remaining 246 proteins, 97 showed a strong correlation with SCA, exceeding the cutoff (|Spearman's correlation coefficient|>0.591). Among these 97 proteins, 44 showed weak or no correlation with lactate levels, and 12 showed weak or no correlation with onset-to-sampling time. Two proteins (AXL receptor tyrosine kinase [AXL] and TIMP Metallopeptidase inhibitor 4 [TIMP-4]) met all the criteria for biomarker extraction (S2 and S3 Tables).

In the GO Slim summary, both proteins were related to the extracellular space in the cellular component category (S1 Fig.). Similarly, in the GO Slim summary for 97 proteins strongly correlated with sudden cardiac arrest, the top cellular component category was also related to the extracellular space (S2 Fig.). The distribution of extracted proteins according to SCA was plotted in Fig 1. Both proteins had higher NPX levels in the SCA group compared to the control group (both p < 0.001). In the SCA group, AXL's NPX values ranged from 7.7 to 10, while TIMP-4's NPX values ranged from 2.6 to 4.9, with the difference between the maximum and minimum values being less than an NPX of 3 for both. The NPX values of the two proteins collected from each center are presented in S3 Fig. The results of hierarchical cluster analysis were plotted using a heatmap. The heatmap also showed that the levels of the two proteins were higher in patients than in the controls. Examining the largest cluster in the heatmap by group, 25 (59.5%) in the SCA group and 22 (52.3%) in the control group were in the same cluster (Fig 2). In the exploratory analysis, the values of these two proteins tended to be higher in patients than in controls for all age groups and BNP levels (Fig 3). In the case of BNP, the overall level was lower in controls than in patients, but the level of the two proteins was higher in patients than in controls when the BNP level was low in both groups (Fig 3).

The AUROC (95% confidence interval [CI]) of AXL model and TIMP-4 model were 0.893 (0.820–0.967) and 0.867 (0.792–0.942), respectively. The predictive performance of the AXL model and the TIMP-4 model was similar (p for comparison = 0.593). However, the NPX values of AXL and TIMP-4 did not showed a strong correlation (correlation coefficient [95% CI]: 0.543 [0.372–0.679]), and when both AXL and TIMP-4 were included in the model, the predictive performance was significantly higher than that of each extracted protein model (AUROC [95% CI] 0.944 [0.895–0.994] for AXL with TIMP-4 model (p for comparison = 0.026 for AXL model and p for comparison = 0.031 for TIMP-4 model).

The AUROC of the baseline model using six traditional risk factors was 0.692 (95% Confidence interval [CI], 0.578–0.806). The addition of AXL, TIMP-4 or both showed a significantly higher predictive power compared to baseline model (AUROC [95% CI] 0.891 [0.817–0.964] for baseline with AXL model and 0.910 [0.910–0.997] for baseline with TIMP-4 model, and 0.954 [0.910–0.997] for baseline with AXL and TIMP-4 model, respectively, all *p* < 0.01 comparison to the baseline model). When both proteins were added to the model, there was a significant difference in AUROC compared to the baseline with AXL model (*p* = 0.007), but there was no significant difference in AUROC compared to the baseline with TIMP-4 model (*p* = 0.121) (Fig 4).

The AUROC of the BNP model was 0.787 (95% CI, 0.688–0.885). While the addition of six traditional risk factors to BNP did not significantly enhance predictive power (AUROC [95% CI] 0.788 [0.689–0.888] for BNP with six traditional risk factors model, p for

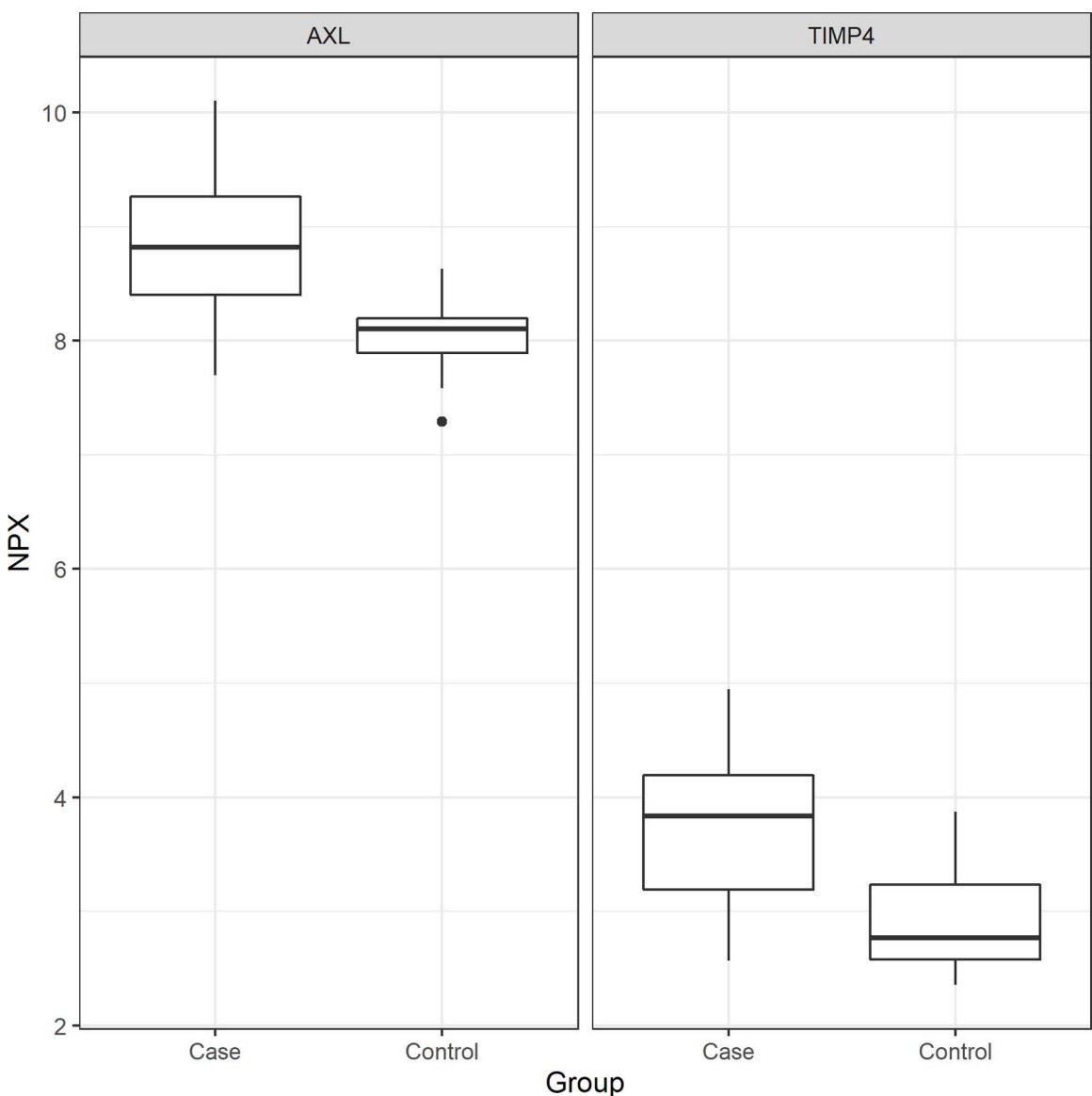

**Fig 1. Boxplot of extracted proteins by group.** AXL, AXL receptor tyrosine kinase, and TIMP4, Metallopeptidase Inhibitor 4.

comparison = 0.072), the inclusion of AXL or TIMP–4 significantly improved the predictive performance compared to the BNP model (AUROC [95% CI] 0.918 [0.853–0.983] and p for comparison = 0.029 for BNP with AXL model and 0.914 [0.850 + 0.978] and p for comparison = 0.005 for BNP with TIMP-4 model) (Fig 5).

## Discussion

In this exploratory study, differences in serum protein profiles of 42 SCA cases with medical causes, aged 20 to 65 years, and whose initial rhythm was shockable on admission to the ED, compared to 42 community-based age- and sex-matched controls, were evaluated using a PEA protein assay. Among 246 proteins that met the quality criteria, 97 showed a strong correlation, satisfying sufficient power in this study's sample size. When extracting proteins unlikely

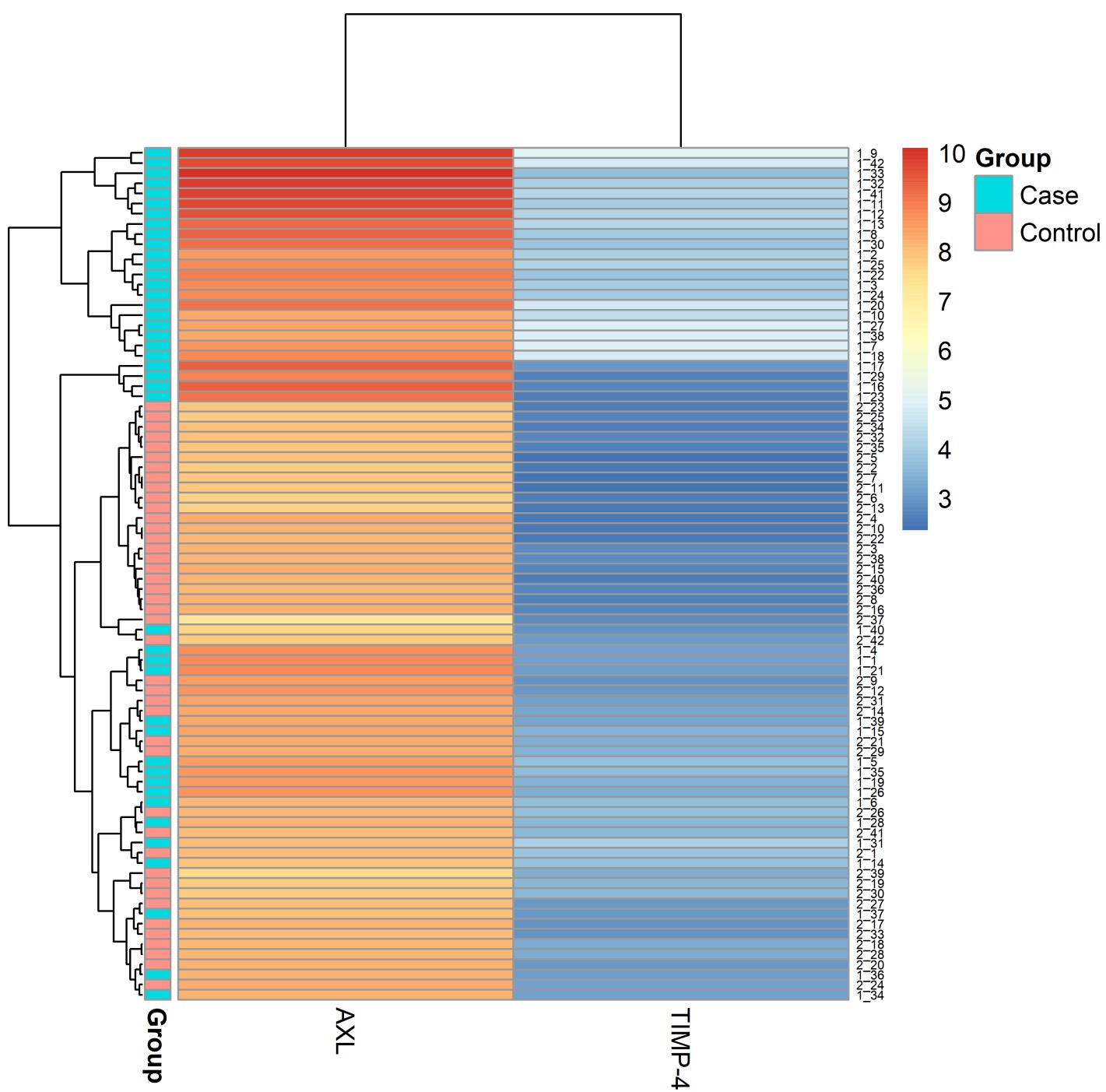

**Fig 2. Heatmap of extracted proteins.** The blue group belongs to cases and the red group belongs to controls. A blue-to-white-to-red end gradient was used as the color scheme based on NPX values. For AXL, NPX values ranged from 7.7 to 10, predominantly appearing in red shades. In contrast, TIMP-4 values ranged from 2.7 to 4.9, primarily appearing in blue shades. The intensity of the red color indicates higher values, while lighter blue represents higher values within the blue range. AXL, AXL receptor tyrosine kinase, and TIMP4, Metallopeptidase Inhibitor 4.

to show post-cardiac arrest changes based on their levels in relation to lactate and sampling time, two proteins (AXL and TIMP-4) were identified. Both proteins demonstrated enhanced discrimination power when added to traditional risk factors in multivariable analysis.

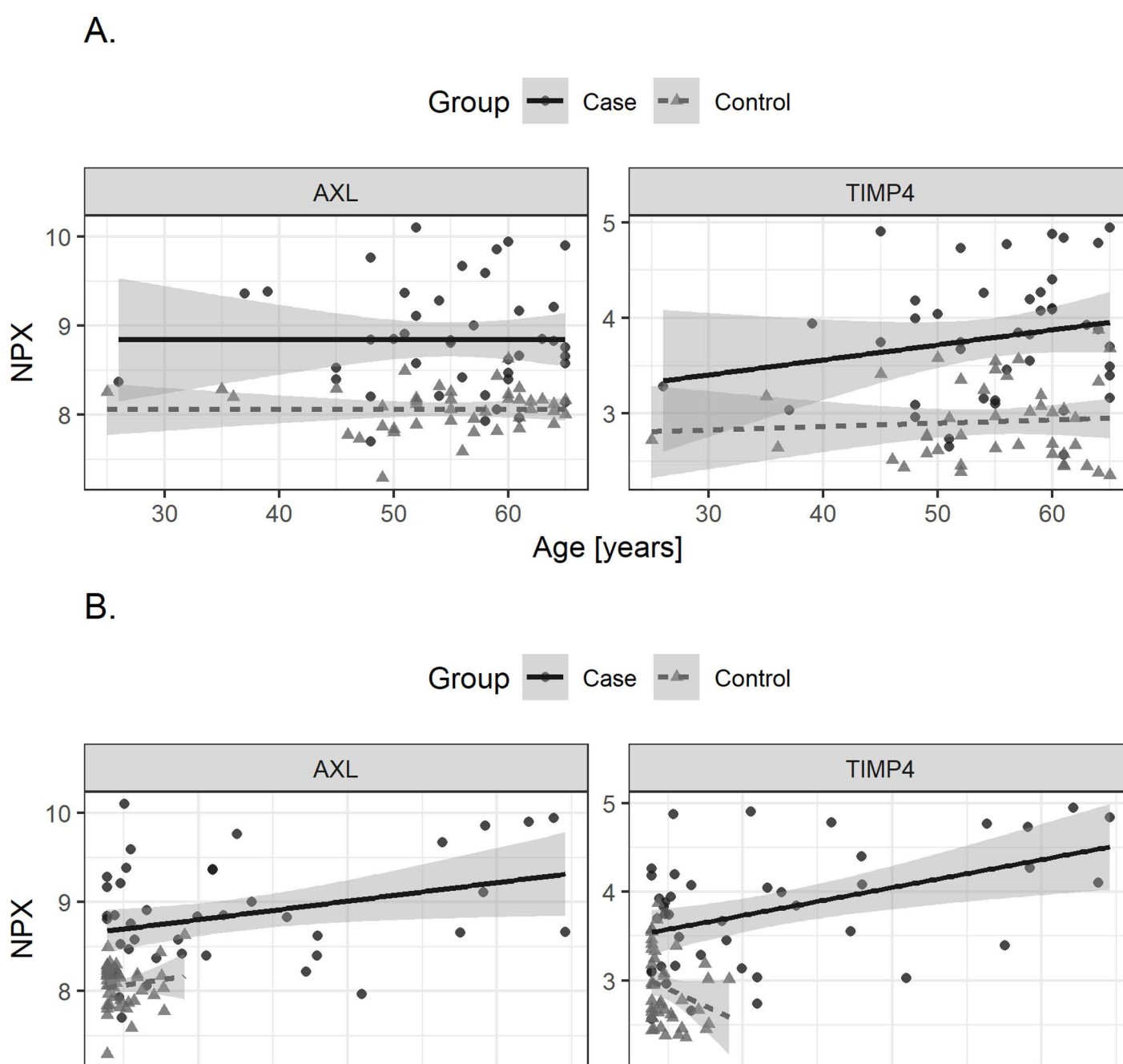

**Fig 3. Scatterplot with fitted linear line between extracted protein levels with A) age, and B) BNP.** Cases and controls were analyzed separately. AXL, AXL receptor tyrosine kinase, BNP, Brain natriuretic peptide, and TIMP4, Metallopeptidase Inhibitor 4. * NPX: Normalized protein expression.

AXL is a cell surface receptor that is involved in signal transduction, from the extracellular matrix (ECM) into the cytoplasm, associated with cell proliferation, adhesion, migration and survival. AXL is an inhibitor of the innate immune response, and is associated with a variety of pathological processes including cancer and autoimmune disorders [19]. AXL also drives

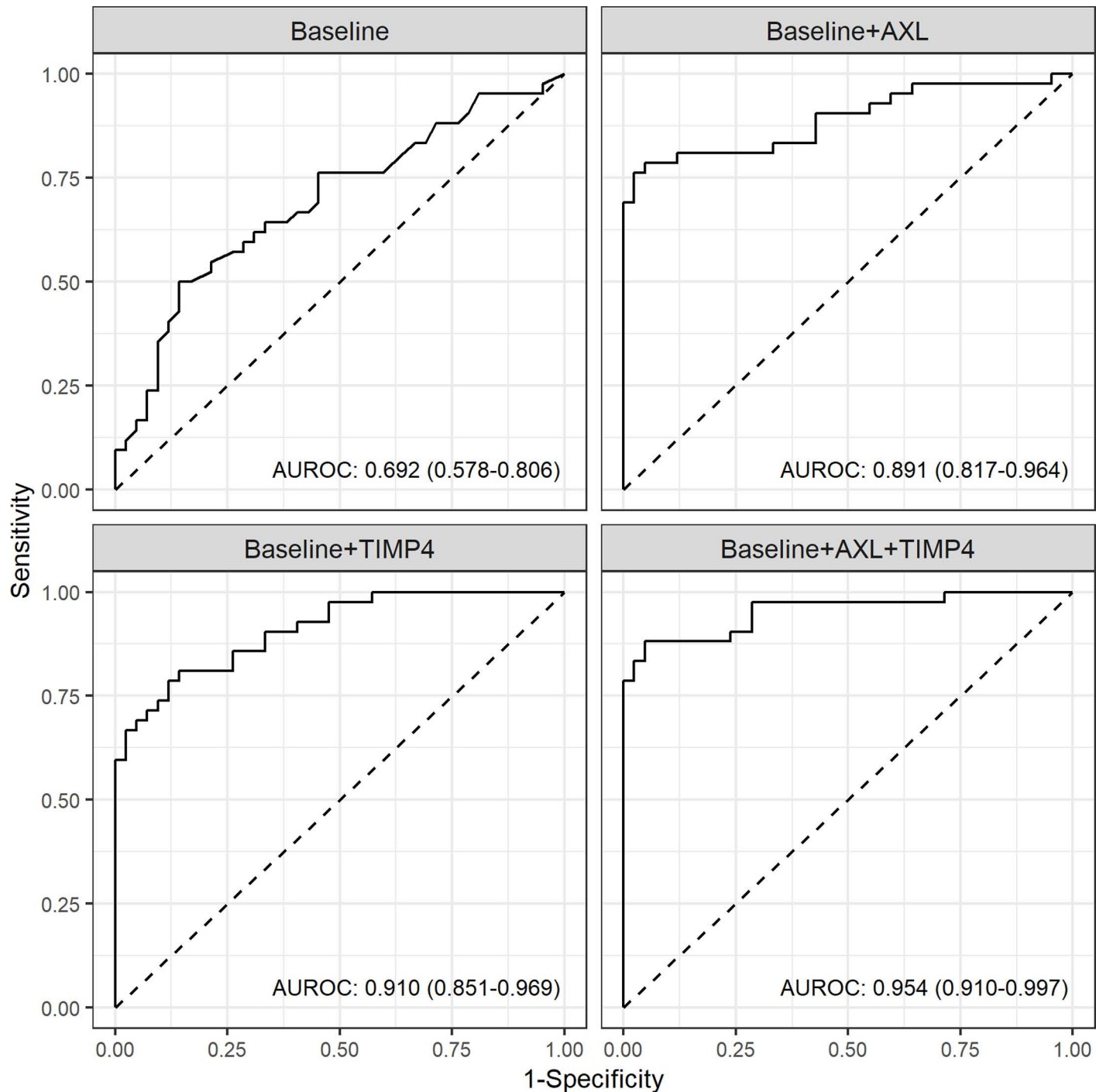

**Fig 4. Area under the receiver operating characteristics curves for multivariable logistic regression models.** Baseline model included 6 traditional risk factors (age, sex, diabetes, hypertension, myocardial infarction, stroke).

cardiac remodelling by regulating endothelial cells, vascular smooth muscle cells, cardiomyocytes, and potentially, fibroblasts [20]. A study using a rat model reported that AXL level increases in the early stages of left ventricular remodelling with pressure overload, with no further increase in heart failure [21].

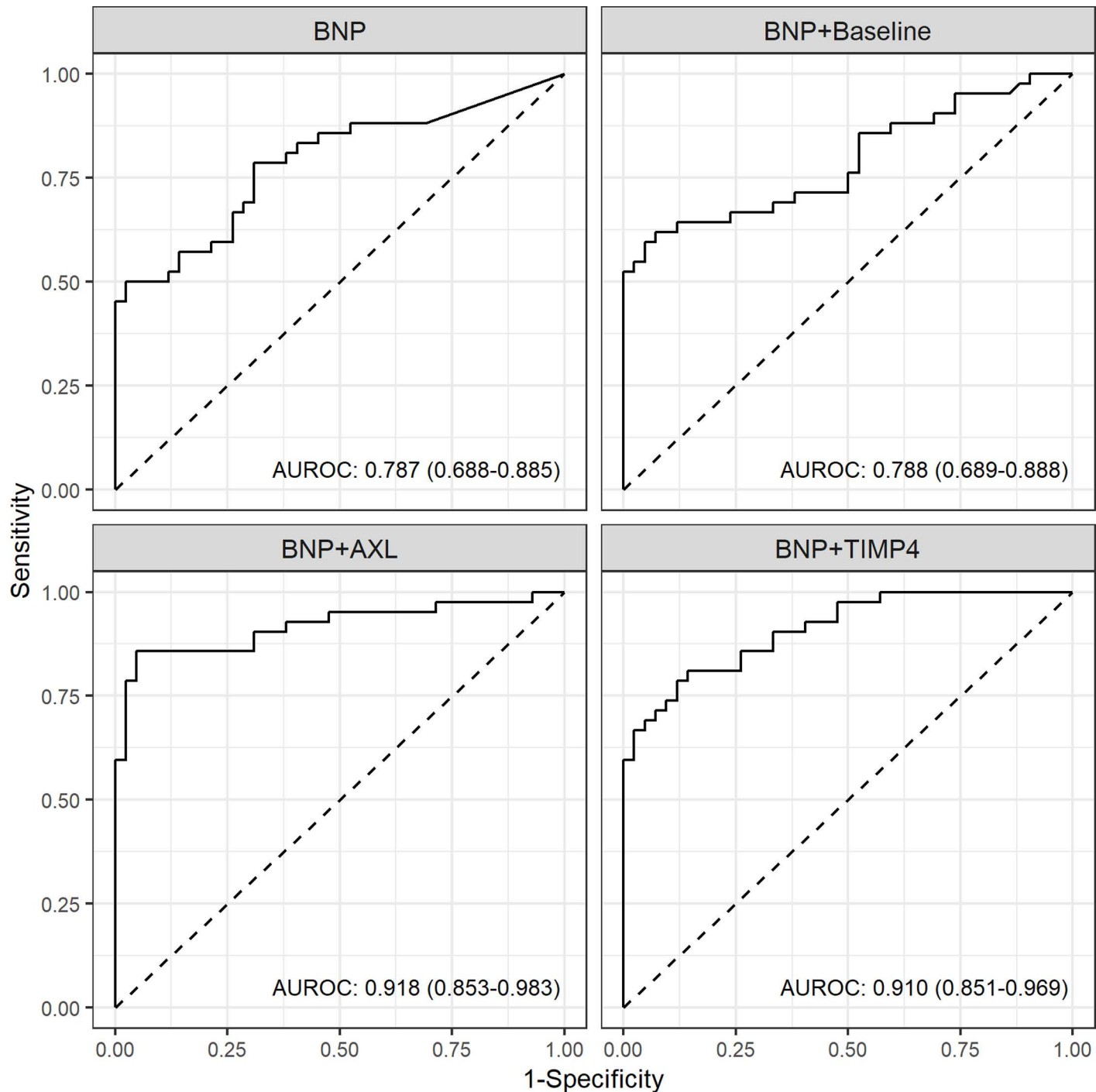

**Fig 5. Area under the receiver operating characteristics curves for multivariable logistic regression models using BNP.** Baseline model included 6 traditional risk factors (age, sex, diabetes, hypertension, myocardial infarction, stroke). BNP, Brain natriuretic peptide.

TIMP-4 inhibits the activity of matrix metalloproteinases (MMP). MMPs play a crucial role in extracellular matrix remodelling and are involved in various physiological processes including tissue development, wound healing, and the malignant conversion of tumour cells [22]. TIMP-4 are the most abundant TIMP protein within the myocardium. A previous study

reported that the TIMP-4 level increased soon after acute myocardial infarction (AMI) and was positively correlated with left ventricular volume changes [23]. In animal model-based studies, an increase in TIMP-4 was observed in compensated left ventricular hypertrophy, but in heart failure, TIMP-4 level or activity had decreased [24,25]. TIMP-4 was also negatively correlated with atrial fibrosis and ECM changes in the atria of rheumatic heart disease with atrial fibrillation [26].

We found that the biomarker analysis results of both proteins were related to the ECM region (S1 Fig.). In addition, we found that all two proteins were directly associated with cardiac remodelling. Cardiac remodelling is one of the main causes of cardiac arrhythmia, ventricular dysfunction, and sudden death [27]. In particular, a previous study has reported an important relationship between cardiac remodelling and arrhythmia, whereby the acquired changes in cardiac structure or function can promote the occurrence of cardiac arrhythmia (arrhythmogenic cardiac remodelling) [28]. The heart can be electrically remodelled by various stimuli in the absence of structural remodelling [29]. Aging itself can cause functional cardiac changes before structural remodelling [30]. In this study, we found that the levels of the two protein biomarkers identified were higher in SCA patients than in controls, under a low BNP level scenario (Fig 3). We also found a significant improvement in predictive performance for SCA when BNP was combined with the extracted protein, compared to the BNP model (Fig 5). These findings suggest that functional or molecular changes in the heart prior to prominent structural changes may affect the risk of cardiac arrest, and that the two biomarkers we discovered might help detect these changes.

A recent study analyzed 330 proteins of 20 SCA survivors and 40 control participants using a TripleTOF® 6600 mass spectrometer with a data-independent acquisition technique, and reported 26 protein biomarkers associated with SCA, of which 20 differentiated SCA from coronary artery disease [8]. In this study, the extracellular matrix was included among the top identified biological processes, which is consistent with our results. This study enhances its validity by conducting additional replication analyses using an additional 29 cases and 57 controls. However, direct comparison with our study is limited because only nine proteins overlap with those analyzed in the current research, and neither AXL nor TIMP-4 were analyzed. In the GO Slim summary for 97 proteins strongly correlated with sudden cardiac arrest, we also found that extracellular components accounted for a significant proportion in the cellular component category. However, the top GO terms in the biological process category were related to inflammatory response and apoptosis-related pathways, while in the molecular function category, the top GO terms were identical protein binding, cytokine activity, and zinc ion binding. This may reflect various pathogenesis mechanisms associated with the occurrence of SCA. Further research targeting various proteins is still necessary.

## Limitations

This study had several limitations. First, the case-control design was used to explore differences in blood test results between SCA patients and controls. Given the unexpected nature of SCA, a case-control design was employed to generate hypotheses more efficiently. Our findings need to be verified in larger cohorts. Second, blood sampling was performed after SCA occurred, which means the samples could be influenced by post-cardiac arrest changes. We reduced this effect by using early post-SCA samples and additional analysis with lactate levels and arrest-to-sampling time. The timing of biomarker measurement is also a concern in previous studies, with samples collected months before or after SCA, leading to interpretation difficulties [8]. We minimized the temporal gap to SCA, but retrospective proximity may still influence data, requiring caution. In addition, single-time sampling without repeats

limited the identification of post-cardiac arrest effects. Third, only pre-specified proteins were analyzed, excluding other known or unknown proteins. Fourth, only Korean patients with shockable rhythm aged ≤65 years were included, requiring caution in interpreting and applying the results. Fifth, AXL and TIMP-4 may be influenced by confounding effects due to their association with other conditions related to SCA. While we included diabetes, hypertension, myocardial infarction, and stroke in our multivariable model, the small sample size limited our ability to adequately adjust for other potential comorbidities. Lastly, because this study was conducted with a retrospective design, it was inherently limited to identifying associations among the variables examined. As a result, it is not possible to draw definitive conclusions regarding causal relationships from the findings.

## Conclusions

Using blood samples from 42 SCA patients and 42 controls, we evaluated the serum levels of 246 proteins, identifying AXL and TIMP-4 as potential SCA biomarkers. Both proteins showed a significant association with SCA and enhanced predictive power with traditional risk factors in multivariable analysis. Our findings suggest that these biomarkers, involved in cardiac remodelling and extracellular matrix processes, may aid early detection and risk assessment of SCA. However, the study's limitations, including its case-control design, single-time sampling, and small sample size, necessitate validation in future studies. Identifying patients with minimal post-cardiac arrest changes could be one potential approach. Selecting patients with witnessed cardiac arrest who have both a short time from arrest to return of spontaneous circulation and a short time to blood sampling could help minimize post-cardiac arrest changes for analysis. Alternatively, analyzing blood samples already collected from cohorts where SCA occurrence is monitored could provide an opportunity to investigate the relationship between identified biomarkers and SCA. Future research should also explore additional biomarkers and verify AXL and TIMP-4's utility in diverse populations to solidify their clinical role in SCA prevention and management.

## Supporting information

**S1 Table. Institutional Review Board (IRB) Numbers of participating hospitals.**
(DOCX)

**S2 Table. Full list of proteins in the analysis with protein selection criteria.**
(DOCX)

**S3 Table. Full list of proteins in the analysis with correlation coefficients at each criteria.**
(DOCX)

**S1 Fig. Goslim summary for biological process, molecular function, and cellular component for AXL and TIMP-4 proteins.** AXL Receptor Tyrosine Kinase; TIMP Metallopeptidase Inhibitor 4.
(DOCX)

**S2 Fig. Goslim summary for biological process, molecular function, and cellular component for 97 proteins with strong correlation with sudden cardiac arrest, exceeding the cutoff (|Spearman's correlation coefficient|>0.516).**
(DOCX)

**S3 Fig. Distribution of AXL and TIMP-4 in patient groups by center.** AXL Receptor Tyrosine Kinase; TIMP Metallopeptidase Inhibitor 4.
(DOCX)

## Acknowledgments

We would like to acknowledge and thank the investigators from all 17 participating university hospitals of the phase II Cardiac Arrest Pursuit Trial with Unique Registry and Epidemiologic Surveillance (CAPTURES-II) project: Sun Young Lee (Seoul National University Hospital), Gyo Jin Ahn (Yonsei University Wonju Severance Christian Hospital), Mi Jin Lee (Kyungpook National University Hospital), Jong-Hak Park (Korea University Ansan Hospital), Su Jin Kim (Korea University Anam Hospital), Sung Bum Oh (Dankook University Hospital), Yong Won Kim (Dongguk University Ilsan Hospital), Jonghwan Shin (Seoul National University Boramae Medical Center), Seung Min Park (Seoul National University Bundang Hospital), Min Seob Sim (Sungkyunkwan University Samsung Medical Center), Won Young Kim (Ulsan University Asan Medical Center), In-Cheol Park (Yonsei University Severance Hospital), Young Hwan Lee (Soonchunhyang University Hospital Bucheon), Hyun Ho Ryu (Chonnam National University Hospital), Sang-Chul Kim (Chungbuk National University Hospital), Ju Ok Park (Hallym University Dongtan Sacred Heart Hospital), and Gu Hyun Kang (Hallym University Kangnam Sacred Heart Hospital).

This study has not been presented at any scientific meeting.

## Author contributions

**Conceptualization:** Jeong Ho Park, Kyoung-Chul Cha, Sung Oh Hwang, Sang Do Shin.

**Formal analysis:** Ha Yeon Shin, Jeong Ho Park, Young Sun Ro, Sang Do Shin.

**Funding acquisition:** Kyoung-Chul Cha.

**Investigation:** Ha Yeon Shin, Woo Jin Jung, Seulki Choi, Ji Hwan Moon, Young Il Roh.

**Methodology:** Ha Yeon Shin, Jeong Ho Park, Hyun Je Kim, Woo Jin Jung, Seulki Choi, Ji Hwan Moon, Young Sun Ro.

**Project administration:** Woo Jin Jung, Young Il Roh, Sung Oh Hwang.

**Resources:** Woo Jin Jung, Young Il Roh, Sung Oh Hwang.

**Software:** Ji Hwan Moon.

**Supervision:** Kyoung-Chul Cha, Hyun Je Kim, Sang Do Shin.

**Validation:** Kyoung-Chul Cha, Seulki Choi, Ji Hwan Moon, Sung Oh Hwang, Sang Do Shin.

**Visualization:** Ha Yeon Shin, Jeong Ho Park.

**Writing – original draft:** Ha Yeon Shin, Jeong Ho Park.

**Writing – review & editing:** Ha Yeon Shin, Jeong Ho Park, Kyoung-Chul Cha, Hyun Je Kim, Woo Jin Jung, Seulki Choi, Young Il Roh, Young Sun Ro, Sung Oh Hwang, Sang Do Shin.

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
