## [Decision Letter · Decision Letter 0]

14 Jan 2025

PONE-D-24-54784Exploratory Study of Serum Protein Biomarkers for Sudden Cardiac Arrest Using Protein Extension Assay: A Case-Control StudyPLOS ONE

Dear Dr. Park,

Thank you for submitting your manuscript to PLOS ONE. After careful consideration, we feel that it has merit but does not fully meet PLOS ONE’s publication criteria as it currently stands. Therefore, we invite you to submit a revised version of the manuscript that addresses the points raised during the review process.

We look forward to receiving your revised manuscript.

Kind regards,

Nanako Kawaguchi

Academic Editor

PLOS ONE

3. Thank you for stating the following financial disclosure:  [This study was supported by the Research Program funded by the Korea Disease Control and Prevention Agency (Grant No: 2017NE3300600, 2017E3300601, 2019P330800, 2021-12-202, 2022-12-204).].  Please state what role the funders took in the study.  If the funders had no role, please state: "The funders had no role in study design, data collection and analysis, decision to publish, or preparation of the manuscript." If this statement is not correct you must amend it as needed. Please include this amended Role of Funder statement in your cover letter; we will change the online submission form on your behalf.

4. Please ensure that you refer to Figure 1 in your text as, if accepted, production will need this reference to link the reader to the figure.

5. Please include a caption for figure 2.

Additional Editor Comments (if provided):

Reviewers' comments:

Reviewer's Responses to Questions

**Comments to the Author**

1. Is the manuscript technically sound, and do the data support the conclusions?

Reviewer #1: Yes

Reviewer #2: Partly

2. Has the statistical analysis been performed appropriately and rigorously? 

Reviewer #1: Yes

Reviewer #2: Yes

3. Have the authors made all data underlying the findings in their manuscript fully available?

Reviewer #1: Yes

Reviewer #2: No

4. Is the manuscript presented in an intelligible fashion and written in standard English?

Reviewer #1: Yes

Reviewer #2: Yes

5. Review Comments to the Author

Reviewer #1: Thank you for the privilege to review this manuscript. The manuscript is well-written and insightful. I do have a few comments which can be found below:

1.I agree with the authors that as this study is retrospective in nature, causal relationships and only associations can be investigated. It is important that they have included this in their limitation.

2.AXL and TIMP-4 are known to be affected in patients with florid cardiac history (eg chronic heart failure). An increased number of sudden cardiac arrest (SCA) cases tend to be observed among these patients. Accordingly, does this confound the findings and observations made in this study?

3.In the limitations, the authors have rightfully included that blood sampling was only performed after SCA occurred, which means that the samples could be influenced by post-cardiac arrest changes. Apart from the interventions that the authors have mentioned in their manuscript, what other interventions can the authors further suggest to improve on this? This is quite a big limitation and I would like the authors to expound more on this.

Reviewer #2: The authors used well-controlled serum samples, defined as those processed within two hours of collection, to minimize the effects of delayed sample processing. Additionally, age- and sex-matched controls were employed to reduce variability from these factors. Furthermore, the authors implemented additional criteria to narrow down candidate protein biomarkers for predicting SCA. Using blood samples from 42 SCA patients and 42 matched controls, they evaluated the serum levels of 246 proteins, identifying AXL and TIMP-4 as potential biomarkers for SCA. Multivariable analysis demonstrated that both proteins could enhance predictive power when combined with traditional risk factors. The authors proposed that these biomarkers, which are involved in cardiac remodeling and extracellular matrix processes, may contribute to the early detection and risk assessment of SCA. However, as authors discussed in the paper, the study has limitations, including small sample size, restriction of quantifiable proteins.

Here are several points the author should address to provide a clearer understanding of the results and make more reliable conclusions.

1. (Page 9, Lines 182-184)

a. Among the patients’ samples, how many samples were obtained from each center? Please include detailed center information either in the main text or in a table.

b. Do the final two proteins (AXL and TIMP-4) show consistent levels across samples from each center? Please address any potential center-specific variation in your analysis.

2. (Page 10, Lines 203-205) Could you address whether delayed blood sample collection affects the levels of the proteins?

3. (Page 11, Lines 216-220) Since the authors used the PEA assay, the following points require clarification:

a. Were the case and control samples randomized before being analyzed using the assay?

b. Please provide more details on how protein levels were normalized. Was it simply log2 scaling? For instance, if we have the NPX values of Protein A and Protein B, can these different protein levels be directly compared using NPX?

c. Were replicate results obtained within the same assay panel to address analytical variation? Additionally, if the number of samples exceeded the 96-plex limit and were processed using separate kits, what normalization strategy was employed to ensure consistency across the kits?

d. The three panels included 18 overlapping proteins. How were these overlapping proteins managed? Could you present how their quantification differed across the panels? If differences were identified, how were these proteins quantified or reconciled using the differing values from the two panels?

4. (Page 12, Lines 232-242) Please include the correlation values for each protein at each extraction step in Supplementary Table 2.

5. (Page 14, Lines 291-292) Are there any sex differences in the levels of the two proteins (AXL and TIMP-4)? Please describe these differences in detail.

6. (Page 14, Lines 293-295) Please provide a detailed explanation of the color scheme used in the heatmap.

a. The color appears to be based on NPX without any normalization. Is there a specific reason for using NPX instead of Z-scores for the heatmap? Using Z-scores for each protein might provide clearer sample clustering.

b. If the color indicates NPX, the two candidate proteins (AXL and TIMP-4) seem to show a positive correlation. Please elaborate on this relationship and provide further discussion regarding its significance.

c. Could you clarify the meaning of the following statement: “Twenty-five of SCA group (59.5%) and 22 of control group (52.3%) were affiliated with the same cluster (Figure 2).” Additional discussion is needed to explain and interpret this finding.

7. (Page 14, Lines 297-299) Since BNP levels are a well-known risk factor for cardiac arrest, how does the BNP’s predictive power differ in case and control?

a. Please address prediction power of these models, if possible.

i. BNP only,

ii. 6 traditional baseline model + BNP

b. Please discuss the predication power of these models, if possible.

i. BNP + AXL

ii. BNP + TIMP-4

iii. BNP + AXL + TIMP-4

iv. 6 traditional baseline model + BNP + AXL

v. 6 traditional baseline model + BNP + TIMP-4

vi. 6 traditional baseline model + BNP + AXL + TIMP-4

8. (Page 14, Lines 300-308)

a. What is the predictive power when using only the two proteins (AXL and TIMP-4) separately or combined, without including the baseline model? Please discuss this in detail.

b. Since the authors used the entire sample to construct the models, there is a high risk of overfitting, particularly due to small sample size. I recommend splitting samples into training set and a test set (e.g., 7:3), constructing the models with the training set, and then validating their predictive power using the test set.

9. (Page 14, Lines 289-290, Page 16, Lines 339-341) For the GO analysis, inputting only two proteins reduces the likelihood of obtaining meaningful insights from common categories. As shown in Supplementary Figure 1, only one category includes both proteins (extracellular space). To draw stronger conclusions from the GO analysis, it would be beneficial to include more proteins. Therefore, I suggest conducting the GO analysis with the 97 proteins that show a strong correlation with SCA. Please provide a more detailed discussion of the results from this expanded analysis.

6. PLOS authors have the option to publish the peer review history of their article (what does this mean? ). If published, this will include your full peer review and any attached files.

**Do you want your identity to be public for this peer review?** For information about this choice, including consent withdrawal, please see our Privacy Policy .

Reviewer #1: **Yes: ** Yoshio Masuda

Reviewer #2: **Yes: ** Jihyeon Lee

---

## [Author Response · Author response to Decision Letter 0]

29 Jan 2025

Authors’ response to the reviewers’ comments:

Submission ID: PONE-D-24-54784

Title: Exploratory Study of Serum Protein Biomarkers for Sudden Cardiac Arrest Using Protein Extension Assay: A Case-Control Study

We would like to thank the reviewers for their valuable comments. We have addressed all the concerns raised by the reviewers and would be willing to make additional changes if needed.

Our major revisions are highlighted in red in the manuscript. Our responses are provided in blue below.

Corresponding author

Jeong Ho Park

RESPONSE: Thank you for your comments. The edits have been made to conform to the PLOS ONE style.

2. Please provide additional details regarding participant consent. In the ethics statement in the Methods and online submission information, please ensure that you have specified (1) whether consent was informed and (2) what type you obtained (for instance, written or verbal, and if verbal, how it was documented and witnessed). If your study included minors, state whether you obtained consent from parents or guardians. If the need for consent was waived by the ethics committee, please include this information. If you are reporting a retrospective study of medical records or archived samples, please ensure that you have discussed whether all data were fully anonymized before you accessed them and/or whether the IRB or ethics committee waived the requirement for informed consent. If patients provided informed written consent to have data from their medical records used in research, please include this information.

RESPONSE: Thank you for your comments. Written informed consent was obtained, and no minors were included in the study.

(REVISON: Ethics Statements

The study was approved by the ethics committees of all participating centers (S1 Table). All participants or their proxy provided written informed consent before taking part in the study and the study complied with the tenets of the Declaration of Helsinki. This study is registered at ClinicalTrials.gov (NCT03700203). No minors were included in the study.

3. Thank you for stating the following financial disclosure: [This study was supported by the Research Program funded by the Korea Disease Control and Prevention Agency (Grant No: 2017NE3300600, 2017E3300601, 2019P330800, 2021-12-202, 2022-12-204).]. Please state what role the funders took in the study. If the funders had no role, please state: "The funders had no role in study design, data collection and analysis, decision to publish, or preparation of the manuscript." If this statement is not correct you must amend it as needed. Please include this amended Role of Funder statement in your cover letter; we will change the online submission form on your behalf.

RESPONSE: Thank you for your comments. We changed financial closure as follows.

(REVISION: Financial support)

This study was supported by the Research Program funded by the Korea Disease Control and Prevention Agency (Grant No: 2017NE3300600, 2017E3300601, 2019P330800, 2021-12-202, 2022-12-204). The funders had no role in study design, data collection and analysis, decision to publish, or preparation of the manuscript

4. Please ensure that you refer to Figure 1 in your text as, if accepted, production will need this reference to link the reader to the figure.

RESPONSE: Thank you for your guidance. We have ensured that Figure 1 is appropriately referenced in the text.

5. Please include a caption for figure 2.

RESPONSE: Thank you for your comments. We added a caption for figure 2.

(REVISION: Figure 2)

Heatmap of extracted proteins. The blue group belongs to cases and the red group belongs to controls. A blue-to-white-to-red end gradient was used as the color scheme based on NPX values. For AXL, NPX values ranged from 7.7 to 10, predominantly appearing in red shades. In contrast, TIMP-4 values ranged from 2.7 to 4.9, primarily appearing in blue shades. The intensity of the red color indicates higher values, while lighter blue represents higher values within the blue range. AXL, AXL receptor tyrosine kinase, and TIMP4, Metallopeptidase Inhibitor 4.

RESPONSE: Thank you for your comments. We reviewed the references and found no retracted papers.

Reviewer #1

Thank you for the privilege to review this manuscript. The manuscript is well-written and insightful. I do have a few comments which can be found below:

1. I agree with the authors that as this study is retrospective in nature, causal relationships and only associations can be investigated. It is important that they have included this in their limitation.

RESPONSE: Thank you for your valuable opinion. We changed sentences in limitation as follows.

(REVISION: Limitation)

Lastly, because this study was conducted with a retrospective design, it was inherently limited to identifying associations among the variables examined. As a result, it is not possible to draw definitive conclusions regarding causal relationships from the findings.

2. AXL and TIMP-4 are known to be affected in patients with florid cardiac history (eg chronic heart failure). An increased number of sudden cardiac arrest (SCA) cases tend to be observed among these patients. Accordingly, does this confound the findings and observations made in this study?

RESPONSE: Thank you for your comments. Although AXL and TIMP-4 can be affected in patients with florid cardiac history, some animal studies reported that AXL and TIMP-4 do not show a significant increase in the state of heart failure. We have described these findings in the discussion section of our study [A study using a rat model reported that AXL level increases in the early stages of left ventricular remodelling with pressure overload, with no further increase in heart failure. / In animal model-based studies, an increase in TIMP-4 was observed in compensated left ventricular hypertrophy, but in heart failure, TIMP-4 level or activity had decreased.] In addition, we also found that the level of the AXL and TIMP-4 was higher in patients than in controls when the BNP level was low in both groups (Figure 2). This finding suggests that AXL and TIMP-4 may be associated with the occurrence of sudden cardiac arrest even in the absence of overt heart failure. However, due to the small sample size, the possibility of a confounding effect could not be entirely evaluated in our study. We have added this point to the limitations section of our study.

(REVISION: Limitation)

Fifth, AXL and TIMP-4 may be influenced by confounding effects due to their association with other conditions related to SCA. While we included diabetes, hypertension, myocardial infarction, and stroke in our multivariable model, the small sample size limited our ability to adequately adjust for other potential comorbidities.

3. In the limitations, the authors have rightfully included that blood sampling was only performed after SCA occurred, which means that the samples could be influenced by post-cardiac arrest changes. Apart from the interventions that the authors have mentioned in their manuscript, what other interventions can the authors further suggest to improve on this? This is quite a big limitation and I would like the authors to expound more on this.

RESPONSE: Thank you for your comments. Identifying patients with minimal post-cardiac arrest changes could be beneficial for future studies. Selecting patients who experienced witnessed cardiac arrest, had a blood sample collected within short time of the event, and had a short CPR duration would allow research to be conducted with reduced post-cardiac arrest changes. Alternatively, analyzing blood samples already collected from cohorts where cardiac arrest occurrence is monitored could provide an opportunity to investigate the relationship between identified biomarkers and cardiac arrest. We plan to pursue additional research in these directions moving forward. We have included this point in the conclusion section.

(REVISION: Conclusion)

However, the study's limitations, including its case-control design, single-time sampling, and small sample size, necessitate validation in future studies. Identifying patients with minimal post-cardiac arrest changes could be one potential approach. Selecting patients with witnessed cardiac arrest who have both a short time from arrest to return of spontaneous circulation and a short time to blood sampling could help minimize post-cardiac arrest changes for analysis. Alternatively, analyzing blood samples already collected from cohorts where SCA occurrence is monitored could provide an opportunity to investigate the relationship between identified biomarkers and SCA.

Reviewer #2

The authors used well-controlled serum samples, defined as those processed within two hours of collection, to minimize the effects of delayed sample processing. Additionally, age- and sex-matched controls were employed to reduce variability from these factors. Furthermore, the authors implemented additional criteria to narrow down candidate protein biomarkers for predicting SCA. Using blood samples from 42 SCA patients and 42 matched controls, they evaluated the serum levels of 246 proteins, identifying AXL and TIMP-4 as potential biomarkers for SCA. Multivariable analysis demonstrated that both proteins could enhance predictive power when combined with traditional risk factors. The authors proposed that these biomarkers, which are involved in cardiac remodeling and extracellular matrix processes, may contribute to the early detection and risk assessment of SCA. However, as authors discussed in the paper, the study has limitations, including small sample size, restriction of quantifiable proteins.

Here are several points the author should address to provide a clearer understanding of the results and make more reliable conclusions.

1. (Page 9, Lines 182-184)

a. Among the patients’ samples, how many samples were obtained from each center? Please include detailed center information either in the main text or in a table.

RESPONSE: Thank you for your comments. We added sentence with center information in results section.

(REVISION: Results)

Among the 60 SCA patients with a shockable rhythm, a random sample of 42 patients was analyzed with 42 matched controls. For the patient group, 42 cases were collected from 13 centers as follows: 5, 4, 7, 5, 2, 2, 1, 2, 5, 1, 4, 1, and 3 cases, respectively. For the control group, 26 and 10 cases were collected from two centers, respectively.

b. Do the final two proteins (AXL and TIMP-4) show consistent levels across samples from each center? Please address any potential center-specific variation in your analysis.

RESPONSE: Thank you for your comments. In the patient group, AXL showed NPX values ranging from 7.7 to 10.1 across all centers, while TIMP-4 showed values between 2.6 and 4.9 across all centers, indicating relatively small center-specific variation. The following figures show the NPX values of AXL and TIMP-4 for the patient group by center.

We added sentences and a supplementary figure in the Results section.

(REVISION: Results)

The distribution of extracted proteins according to SCA was plotted in Figure 1. Both proteins had higher NPX levels in the SCA group compared to the control group (both p<0.001). In the SCA group, AXL's NPX values ranged from 7.7 to 10, while TIMP-4's NPX values ranged from 2.6 to 4.9, with the difference between the maximum and minimum values being less than an NPX of 3 for both. The NPX values of the two proteins collected from each center are presented in S3 Fig.

2. (Page 10, Lines 203-205) Could you address whether delayed blood sample collection affects the levels of the proteins?

RESPONSE: Thank you for your comments. Sudden cardiac arrest induces systemic hypoxia and inflammation, which can affect various protein levels in delayed blood samples. However, due to the nature of this study being conducted with single samples, it was not possible to distinguish whether the observed differences in protein levels were due to delayed sampling time or inherent differences in protein levels. To address this, only proteins with low correlation between onset-to-sampling time and protein levels were selected for analysis. However, this analysis could not completely eliminate the effects of sampling time. This limitation of the study has been described in the limitations section as follows.

[Second, blood sampling was performed after SCA occurred, which means the samples could be influenced by post-cardiac arrest changes. We reduced this effect by using early post-SCA samples and additional analysis with lactate levels and arrest-to-sampling time. The timing of biomarker measurement is also a concern in previous studies, with samples collected months before or after SCA, leading to interpretation difficulties.[8] We minimized the temporal gap to SCA, but retrospective proximity may still influence data, requiring caution. In addition, single-time sampling without repeats limited the identification of post-cardiac arrest effects.]

3. (Page 11, Lines 216-220) Since the authors used the PEA assay, the following points require clarification:

a. Were the case and control samples randomized before being analyzed using the assay?

RESPONSE: Thank you for your comments. Since the samples were randomized before analysis, the placement of case and control samples within the plate was random. We added sentences in methods section.

(REVISION: Methods-Protein analysis) Since the samples were randomized before analysis, the placement of case and control samples within the plate was random.

b. Please provide more details on how protein levels were normalized. Was it simply log2 scaling? For instance, if we have the NPX values of Protein A and Protein B, can these different protein levels be directly compared using NPX?

RESPONSE: Thank you for your comments. NPX is a relative quantification unit logarithmically related to protein concentration. Even if two different proteins have the same NPX values, their absolute concentrations may differ. However, through inter-plate normalization, the NPX of the same protein across different plates can be compared equivalently. This is done by using inter-plate controls (IPC), which help adjust for any systematic differences, allowing for accurate and consistent comparisons of protein levels across different plates. We revised methods section as follows.

(REVISION: Methods-Protein analysis)

Protein levels were measured on a relative scale and presented as normalised protein expression (NPX), which is an arbitrary unit on a log2 scale. A high NPX value corresponds to a high protein concentration. The levels of different proteins cannot be compared using NPX. However, using inter-plate controls (IPC), any systematic differences across different plates were adjusted; therefore, a consistent comparison of the same protein levels across different plates was possible. The IPC consists of a pool of 92 antibodies, each with unique DNA-tags, and is included in triplicate on each plate. The IPC serves as a synthetic sample, expected to give a high signal across all assays, and the median of the IPC triplicates is used to normalize each assay, correcting for potential variation between runs and pla

---

## [Decision Letter · Decision Letter 1]

4 Feb 2025

Exploratory Study of Serum Protein Biomarkers for Sudden Cardiac Arrest Using Protein Extension Assay: A Case-Control Study

PONE-D-24-54784R1

Dear Dr. Park,

We’re pleased to inform you that your manuscript has been judged scientifically suitable for publication and will be formally accepted for publication once it meets all outstanding technical requirements.

Kind regards,

Nanako Kawaguchi

Academic Editor

PLOS ONE

---

## [Editor Report · Acceptance letter]

PONE-D-24-54784R1

PLOS ONE

Dear Dr. Park,

I'm pleased to inform you that your manuscript has been deemed suitable for publication in PLOS ONE. Congratulations! Your manuscript is now being handed over to our production team.

Kind regards,

on behalf of

Dr. Nanako Kawaguchi

Academic Editor

PLOS ONE